# PLANCKIAN JITTER: COUNTERING THE COLOR-CRIPPLING EFFECTS OF COLOR JITTER ON SELF-SUPERVISED TRAINING

**Simone Zini**[1]*, **Alex Gomez-Villa**[2], **Marco Buzzelli**[1], **Bartłomiej Twardowski**[2,4],
**Andrew D. Bagdanov**[3], **Joost van de Weijer**[2]

[1]Department of Informatics Systems and Communication, University of Milano – Bicocca, Milan, Italy
`{simone.zini, marco.buzzelli}@unimib.it`
[2]Computer Vision Center, Universitat Autònoma de Barcelona, Barcelona, Spain
`{agomezvi, btwardowski,joost}@cvc.uab.es`
[3]Media Integration and Communication Center, University of Florence, Florence, Italy
`andrew.bagdanov@unifi.it`
[4]IDEAS NCBR, Warsaw, Poland

## ABSTRACT

Several recent works on self-supervised learning are trained by mapping different augmentations of the same image to the same feature representation. The data augmentations used are of crucial importance to the quality of learned feature representations. In this paper, we analyze how the color jitter traditionally used in data augmentation negatively impacts the quality of the color features in learned feature representations. To address this problem, we propose a more realistic, physics-based color data augmentation – which we call *Planckian Jitter* – that creates realistic variations in chromaticity and produces a model robust to illumination changes that can be commonly observed in real life, while maintaining the ability to discriminate image content based on color information. Experiments confirm that such a representation is complementary to the representations learned with the currently-used color jitter augmentation and that a simple concatenation leads to significant performance gains on a wide range of downstream datasets. In addition, we present a color sensitivity analysis that documents the impact of different training methods on model neurons and shows that the performance of the learned features is robust with respect to illuminant variations. Official code available at: `https://github.com/TheZino/PlanckianJitter`

## 1 INTRODUCTION

Self-supervised learning enables the learning of representations without the need for labeled data (Doersch et al., 2015; Dosovitskiy et al., 2014). Several recent works learn representations that are invariant with respect to a set of data augmentations and have obtained spectacular results (Grill et al., 2020; Chen & He, 2021; Caron et al., 2020), significantly narrowing the gap with supervised learned representations. These works vary in their architectures, learning objectives, and optimization strategies, however they are similar in applying a common set of data augmentations to generate different image views. These algorithms, while learning to map these different views to the same latent representation, learn rich semantic representations for visual data. The set of transformations (data augmentations) used induces invariances that characterize the learned visual representation.

Before deep learning revolutionized the way visual representations are learned, features were hand-crafted to represent various properties, leading to research on shape (Lowe, 2004), texture (Manjunath & Ma, 1996), and color features (Finlayson & Schaefer, 2001; Geusebroek et al., 2001). Color features were typically designed to be invariant to a set of scene-accidental events such as shadows, shading, and illuminant and viewpoint changes. With the rise of deep learning, feature

---

*Corresponding author

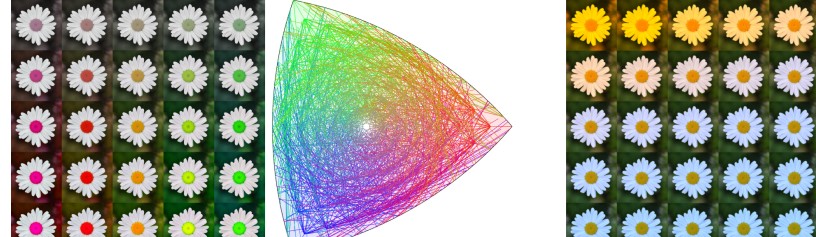

Figure 1: Default color jitter (left) and Planckian Jitter (right). Augmentations based on default color jitter lead to unrealistic images, while Planckian Jitter leads to a set of realistic ones. The ARC chromaticity diagrams for each type of jitter are computed by sampling initial RGB values and mapping them into the range of possible outputs given by each augmentation. These diagrams show that Planckian Jitter transforms colors along chromaticity lines occurring in nature when changing the illuminant, whereas default color jitter transfers colors throughout the whole chromaticity plane.

representations that simultaneously exploit color, shape, and texture are learned implicitly and the invariances are a byproduct of end-to-end training (Krizhevsky et al., 2009). Current approaches to self-supervision learn a set of invariances implicitly related to the applied data augmentations.

In this work, we focus on the currently de facto choice for color augmentations. We argue that they seriously cripple the color quality of learned representations and we propose an alternative, physics-based color augmentation. Figure 1 (left) illustrates the currently used color augmentation on a sample image. It is clear that the applied color transformation significantly alters the colors of the original image, both in terms of hue and saturation. This augmentation results in a representation that is invariant with respect to surface reflectance – an invariance beneficial for recognizing classes whose surface reflectance varies significantly, for example many man-made objects such as cars and chairs. However, such invariance is expected to hurt performance on downstream tasks for which color is an important feature, like natural classes such as birds or food. One of the justifications is that without large color changes, mapping images to the same latent representation can be purely done based on color and no complex shape or texture features are learned. However, as a result the quality of the color representation learned with such algorithms is inferior and important information on surface reflectance might be absent. Additionally, some traditional supervised learning methods propose domain-specific variations of color augmentation Galdran et al. (2017); Xiao et al. (2019).

In this paper we propose an alternative color augmentation (Figure 1, right) and we assess its impact on self-supervised learning. We draw on the existing color imaging literature on designing features invariant to illuminant changes commonly encountered in the real world (Finlayson & Schaefer, 2001). Our augmentation, called *Planckian Jitter*, applies physically-realistic illuminant variations. We consider the illuminants described by Planck's Law for black-body radiation, that are known to be similar to illuminants encountered in real-life (Tominaga et al., 1999). The aim of our color augmentation is to allow the representation to contain valuable information about the surface reflectance of objects – a feature that is expected to be important for a wide range of downstream tasks. Combining such a representation with the already high-quality shape and texture representation learned with standard data augmentation leads to a more complete visual descriptor that also describes color.

Our experiments show that self-supervised representations learned with Planckian Jitter are robust to illuminant changes. In addition, depending on the importance of color in the dataset, the proposed Planckian jitter outperforms the default color jitter. Moreover, for all evaluated datasets the combination of features of our new data augmentation with standard color jitter leads to significant performance gains of over 5% on several downstream classification tasks. Finally, we show that Planckian Jitter can be applied to several state-of-the-art self-supervised learning methods.

## 2 BACKGROUND AND RELATED WORK

**Self-supervised learning and contrastive learning.** Recent improvements in self-supervision learn semantically rich feature representations without the need for labelled data. In SimCLR (Chen et al., 2020a) similar samples are created by augmenting an input image, while dissimilar are chosen

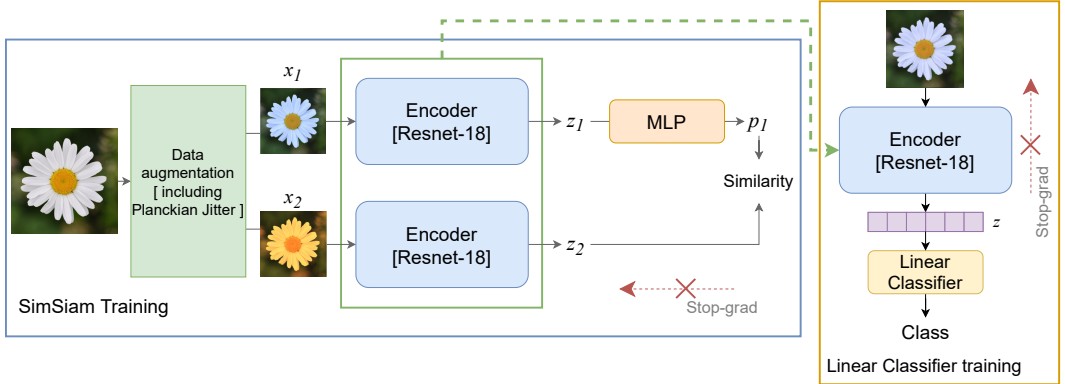

Figure 2: SimSiam training procedure exploiting Planckian-based data augmentation (left), and fine-tuning the linear classifier using the trained encoder (right).

randomly (Chen et al., 2020a). To improve efficiency, MoCo (He et al., 2020) and its enhanced version (Chen et al., 2020b) use a memory bank for learned embeddings which makes sampling efficient. This memory is kept in sync with the rest of the network during training via a momentum encoder. Several methods do not rely on explicit contrastive pairs. BYOL uses an asymmetric network incorporating an additional MLP predictor between the outputs of the two branches (Grill et al., 2020). One of the branches is kept "offline" and is updated by a momentum encoder. SimSiam defines a simplified solution without a momentum encoder (Chen & He, 2021). It obtains similar high-quality results and does not require a large minibatch size, in contrast to other methods.

We use the SimSiam method to verify our proposed color augmentation (we also apply it to Sim-CLR (Chen et al., 2020a) and Barlow Twins (Zbontar et al., 2021) in the experiments). The main component of the network is a CNN-based asymmetric siamese image encoder. One branch has an additional MLP predictor whose output aims to be as close as possible to the other (Figure 2). The second branch is not updated during backpropagation. A negative cosine loss function is used:

$$\mathcal{L} \quad = \quad \frac{1}{2} \left[ \mathcal{D}(p_1, \text{stopgrad}(z_2)) + \mathcal{D}(p_2, \text{stopgrad}(z_1)) \right] \tag{1}$$

$$\mathcal{D}(p_A, z_B) \quad = \quad -\frac{p_A}{\|p_A\|_2} \cdot \frac{z_B}{\|z_B\|_2}, \tag{2}$$

where $z_1, z_2$ are representations for two different augmented versions, $x_1$ and $x_2$, of the same image $x$. The MLP is applied in alternation to either $z_1$ or $z_2$, producing respectively $p_1$ or $p_2$. Note that Figure 2 only shows an instance for $x_1$ and does not show $p_2$. The stopgrad($\cdot$) operation blocks the gradient during the backpropagation. In SimSiam no contrastive term is used and only similarity is enforced during learning.

**Data augmentation.** Data augmentation plays a central role in the self-supervised learning process. Authors Chen et al. (2020a) and Zbontar et al. (2021) discuss the importance of the different data augmentations. A set of well-defined transformations was proposed for SimCLR (Chen et al., 2020a). This set is commonly accepted and used in several later works. The augmentations include: rotation, cutout, flip, color jitter, blur and Grayscale. These operations are randomly applied to an image to generate the different views $x_1$, $x_2$ from which are extracted the features $z_1$ and $z_2$ used in the self-supervision loss in Eq. 2. Applied to the same image, contrastive-like self-supervision learns representations invariant to such distortions.

This multiple view creation is task-related (Tian et al., 2020), however color jittering operating on hue, saturation, brightness and contrast, is one of the most important ones in terms of overall usefulness of the learned representation for downstream tasks (Chen et al., 2020a; Zbontar et al., 2021). Color jitter induces a certain level of color invariance (invariance to hue, saturation, brightnesss and contrast) which are consequently transferred to the downstream task. As a consequence, we expect these learned features to underperform on downstream tasks for which color is crucial. Xiao et al. (2020) were the first to point out that the imposed invariances might not be beneficial for downstream tasks. As a solution, they propose to learn different embedding spaces in parallel that capture each of

the invariances. Differently than them, we focus on the color distortion and propose a physics-based color augmentation that allows learning invariance to physically realistic color variations.

Color imaging has a long tradition in research on color features invariant to scene-accidental events such as shading, shadows, and illuminant changes (Geusebroek et al., 2001; Finlayson & Schaefer, 2001). Invariant features were found to be extremely beneficial for object recognition. The invariance to hue and saturation changes induced by color jitter, however, is detrimental to object recognition for classes in which color characteristics are fundamentally discriminative. Therefore, in this work we revisit early theory on illuminant invariance (Finlayson & Schaefer, 2001) to design an improved color augmentation that induces invariances common in the real world and that, when used during self-supervised learning, does not damage the color quality of the learned features.

## 3 METHODOLOGY

The image transformations introduced by default color jitter creates variability in training data that indiscriminately explores all hues at various levels of saturation. The resulting invariance is useful for downstream tasks where chromatic variations are indeed irrelevant (e.g. car color in vehicle recognition), but is detrimental to downstream tasks where color information is critical (e.g. natural classes like birds and vegetables). The main motivation for applying strong color augmentations is that this it leads to very strong shape and texture representations. Indiscriminately augmenting color information in the image requires that the representation solve the matching problem using shape (Chen et al., 2020a)[1].

As an alternative to color jitter, we propose a physics-based color augmentation that mimics color variations due to illuminant changes commonly encountered in the real world. The aim is to reach a representation that does not have the color crippling effects of color jitter, and that better describes classes for which surface reflectance is a determining feature. When combined with default color jitter, this representation should also provide a high-quality shape/texture and color representation.

### 3.1 PLANCKIAN JITTER

We call our color data augmentation procedure *Planckian Jitter* because it exploits the physical description of a black-body radiator to re-illuminate training images within a realistic illuminant distribution (Finlayson & Schaefer, 2001; Tominaga et al., 1999). The resulting augmentations are more realistic than those of the default color jitter (see Fig. 1). The resulting learned, self-supervised feature representation is thus expected to be robust to illumination changes commonly observed in real-world images, while simultaneously maintaining the ability to discriminate the image content based on color information.

Given an input RGB training image $I$, our Planckian Jitter procedure applies a chromatic adaptation transform that simulates realistic variations in the illumination conditions. The data augmentation procedure is as follows:

1. we sample a new illuminant spectrum $\sigma_T(\lambda)$ from the distribution of a black-body radiator;
2. we transform the sampled spectrum $\sigma_T(\lambda)$ into its sRGB representation $\rho_T \in \mathbb{R}^3$;
3. we create a jittered image $I'$ by reilluminating $I$ with the sampled illuminant $\rho_T$; and
4. we introduce brightness and contrast variation, producing a Planckian-jittered image $I''$.

A radiating black body at temperature $T$ can be synthesized using Planck's Law (Andrews, 2010):

$$\sigma_T(\lambda) = \frac{2\pi hc^2}{\lambda^5(e^{\frac{hc}{kT\lambda}} - 1)} \text{ W/m}^3, \tag{3}$$

where $c = 2.99792458 \times 10^8$ m/s is the speed of light, $h = 6.626176 \times 10^{-34}$ Js is Planck's constant, and $k = 1.380662 \times 10^{-23}$ J/K is Boltzmann's constant. We sampled $T$ in the interval between $3000K$ and $15000K$ which is known to result in a set of illuminants that can be encountered in real life (Tominaga et al., 1999). Then, we discretized wavelength $\lambda$ in 10nm steps ($\Delta\lambda$) in the interval between 400nm and 700nm. The resulting spectra are visualized in Figure 4 (left) in Appendix A.1.

---

[1]This is pointed out in the discussion of Figure 5 in Chen et al. (2020a)

The conversion from spectrum into sRGB is obtained according to Wyszecki & Stiles (1982):

1. we first map the spectrum into the corresponding XYZ stimuli, using the 1931 CIE standard observer color matching functions $c^{\{X,Y,Z\}}(\lambda)$, in order to bring the illuminant into a standard color space that represents a person with average eyesight;

2. we normalize this tristimulus by its $Y$ component, convert it into the CIE 1976 L*a*b color space, and fix its L component to 50 in a 0-to-100 scale, allowing us to constrain the intensity of the represented illuminant in a controlled manner as a separate task; and

3. we then convert the resulting values to sRGB, applying a gamma correction and obtaining $\rho_T = \{R, G, B\}$; the resulting distribution of illuminants is visualized with the Angle-Retaining Chromaticity diagram (Buzzelli et al., 2020) in Figure 4 (right) in Appendix A.1.

All color space conversions assume a D65 reference white, which means that a neutral surface illuminated by average daylight conditions would appear achromatic. Once the new illuminant has been converted in sRGB, it is applied to the input image $I$ by resorting to a Von-Kries-like transform (von Kries, 1902) given by the following channel-wise scalar multiplication:

$$I'^{\{R,G,B\}} = I^{\{R,G,B\}} \cdot \{R, G, B\}/\{1, 1, 1\}, \tag{4}$$

where we assume the original scene illuminant to be white (1,1,1). Finally, brightness and contrast perturbations are introduced to simulate variations in the intensity of the scene illumination:

$$I'' = c_B \cdot c_C \cdot I' + (1 - c_C) \cdot \mu\left(c_B \cdot I'\right), \tag{5}$$

where $c_B = 0.8$ and $c_C = 0.8$ represent, respectively, brightness and contrast coefficients, and $\mu$ is a spatial average function.

## 3.2 COMPLIMENTARITY OF SHAPE, TEXTURE AND COLOR REPRESENTATIONS

The self-supervised learning paradigm involves a pretraining phase that relies on data augmentation to produce a set of features with certain invariance properties. These features are then used as the representation for a second phase, where we learn a given supervised downstream task. The default color jitter augmentation generates features that are strongly invariant to color information, resulting in high-quality representations of shape and texture, but that is an inferior descriptor of surface reflectances (i.e. the color of objects). Our augmentation based on Planckian Jitter (see Figure 1) is based on transformations mimicking the physical color variations in the real world due to illuminant changes. As a result, the learned representation yields a high-quality color description of scene objects (this is also verified in Appendix A.8). However, it likely leads to a drop in the quality of the shape and texture representation (since color can be used to solve cases where previously shape/texture were required). To exploit the complimentarity of the two representations, we propose to learn both – one with color jitter and one with Planckian Jitter – and to then concatenate the results in a single representation vector (of 1024 dimensions, i.e. twice the original size of 512). We call this *Latent space combination (LSC)*.

## 4 EXPERIMENTAL RESULTS

In this section, we analyze the color sensitivity of the learned backbone networks, verify the superiority of the proposed color data augmentation method compared to the default color jitter on color datasets, and evaluate the impact on downstream classification tasks. We report additional results on computational time of the proposed Planckian augmentation in Appendix A.5.

## 4.1 TRAINING AND EVALUATION SETUP

We perform unsupervised training on two datasets: CIFAR-100 (Krizhevsky et al., 2009) ($32 \times 32$) and ImageNet ($224 \times 224$). [2] We slightly modify the ResNet18 architecture to accommodate $32 \times 32$ images: the kernel size of the first convolutional was reduced from $7 \times 7$ to $3 \times 3$ and the first max

---

[2]We conduct the investigative part of our research in an agile manner on low-resolution images, then transfer the most significant configurations to a higher-resolution, to ether confirm or refute the initial hypotheses.

pooling layer was removed. SimSiam training was performed using Stochastic Gradient Descent with a starting learning rate of $0.03$, a cosine annealing learning rate scheduler, and mini-batch size of $512$ (as in original SimSiam work by Chen & He (2021)). For the training on the 1000-class ImageNet training set, we follow the same procedure as Chen & He (2021) with ResNet50.

The linear classifier training at resolution $32 \times 32$ was performed on CIFAR-100 and FLOWERS-102 (Nilsback & Zisserman, 2008). CIFAR-100 is used as a baseline for the classification task. The linear classifier training for CIFAR-100 is done with Stochastic Gradient Descent for 500 epochs with a starting learning rate $0.1$, a cosine annealing learning rate scheduler, and mini-batch size of $512$. The FLOWERS-102 dataset with 102 classes was selected to assess the quality of the features extracted in scenarios where color information plays an important role. Images from FLOWERS-102 are resized to $32 \times 32$ pixels to match the input dimensions of the pretrained model. Here we used the Adam optimizer with initial learning rate of $0.03$.

For training linear classifiers at resolution $224 \times 224$ for downstream tasks we follow the evaluation protocol of Chen & He (2021). We use six different datasets: IMAGENET, FLOWERS-102, the fine-grained VEGFRU (Saihui Hou & Wang, 2017), CUB-200 (Welinder et al., 2010), T1K+ (Cusano et al., 2021), and USED (Ahmad et al., 2016), all resized to $224 \times 224$ pixels. More details about the datasets are provided in Appendix A.2. In the case of CUB-200, each image was cropped using the bounding boxes given in the dataset annotations. For T1K+, we used the 266 class labeling to train and test the linear classifier.

To assess the impact of color data augmentations we define six different configurations:

- *Default Color Jitter (CJ):* the default configuration, as used in SimSiam and SimCLR, uses both Random Color Jitter and Random Grayscale operations.
- *Default Color Jitter w/o Grayscale (CJ-)*: same as *Default*, without Random Grayscale.
- *Planckian Jitter (PJ)*: uses the complete proposed Planckian Jitter operating on chromaticy, brightness, and contrast aspects of the images. No Random Grayscale is applied.
- *LSC Default Color Jitter + Planckian Jitter ([CJ,PJ]*: This latent space combination (simple concatenation of representations) combines the default color jitter with our Planckian jitter. It allows evaluation of the complimentary nature of the representations.
- *LSC Default Color Jitter + Default Color Jitter w/o Grayscale ([CJ,CJ-])*: We combine the default color jitter with a version without the Grayscale augmentation, since this representation is also expected to result in a better color representation.
- *LSC of two Default Color Jitter Models ([CJ,CJ])*: We also show results of simply concatenating two independently trained models (trained from different seeds) with default color jitter (an ensemble of two models).

In all experiments, these are combined with the other default augmentations (crop, flip, and blur).

## 4.2 COLOR SENSITIVITY ANALYSIS

We perform a robustness analysis on the VegFru and CUB-200 datasets with realistic illuminant variations, and analyzed sensitivity to color information. This experiment is driven by two motivations: to verify that we obtain invariance to the transformation applied during training, and to characterize the degradation of different non-Planckian training modalities. We assume as reference point the D65 illuminant, which for the purpose of this test is considered the default illuminant in every image. Given the different backbones pretrained on IMAGENET, we then train a linear classifier on this dataset (assumed to be under white illumination). For testing we create different versions of the dataset, each illuminated by illuminants of differing color temperature. This allows us to evaluate the robustness of the learned representations with respect to these illuminant changes.

Results are given in Figure 3(a) (more results are provided in Figure 7 from Appendix A.4). *Planckian Jitter* obtains a remarkably stable performance between 4000K and 14000K, while *Default Color Jitter* is more sensitive to the illumination color and the classification accuracy decreases when the scene illuminant moves away from white. We also see that the combination of default and Planckian Jitter obtains the best results for all illuminants and manages to maintain a high-level of invariance with respect to the illuminant color. Among the non-Planckian curves, default color jitter (CJ) is the most invariant, followed by CJ+CJ- (although showing better performance at D65), and finally CJ-.

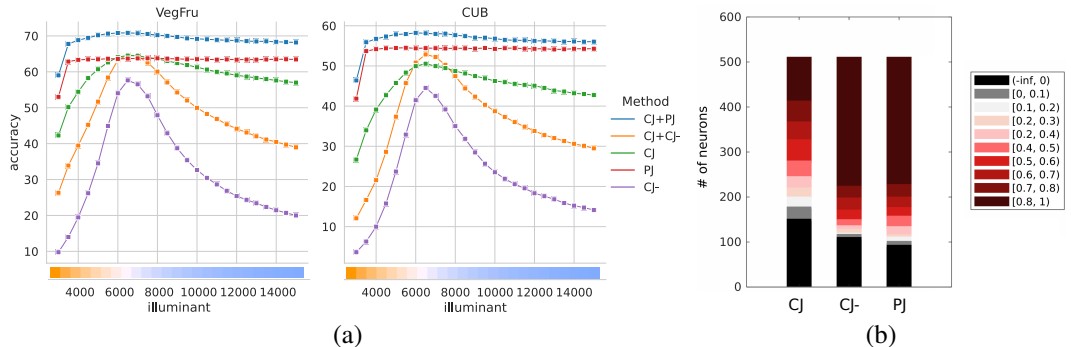

(a)                                                                    (b)

Figure 3: Color sensitivity analysis. (a) Robustness to illuminant change: we report the accuracies by differently-trained backbones as a function of illuminant. (b) The color sensitivity indexes computed for the different configurations used for training the backbone.

Table 1: Accuracy results with ablation on color augmentations. Self-supervised training is performed on CIFAR-100 and the learned features are evaluated at $(32 \times 32)$ on CIFAR-100 and FLOWERS-102. Augmentation techniques include variations in hue and saturation (H&S), brightness and contrast (B&C), Planckian-based chromaticity (P), and random Grayscale conversions (G). Accuracy refers to the linear classifiers trained with features extracted from the different backbones.

| AUGMENTATION | H&S | B&C | G | P | CIFAR-100 | FLOWERS-102 |
|---|---|---|---|---|---|---|
| None | | | | | 41.93% | 36.47% |
| Default Color Jitter | ✓ | ✓ | ✓ | | 59.93% | 30.00% |
| | ✓ | ✓ | | | 41.96% | 36.96% |
| | ✓ | | | | 32.46% | 39.11% |
| | | | | ✓ | 36.10% | 39.51% |
| | | ✓ | | | 31.78% | 41.96% |
| Planckian Jitter | | ✓ | | ✓ | 47.31% | 42.75% |

In order to understand the impact of the color information on each neuron in trained models, we conducted an analysis using the color selectivity index described by Rafegas & Vanrell (2018). This index measures neuron activation when color is present or absent in input images. We computed the index for the last layer of different backbones, and high values indicate color-sensitive neurons. See Appendix A.3 for more details on color selectivity. The results are shown in Figure 3(b) and indicate the number of color-sensitive neurons for each of the considered models. It is clear that the default color jitter has far fewer neurons dedicated to color description. This result confirms the hypothesis that models trained in this way are color invariant, a property that negatively affects the model in scenarios where color information has an important role as seen in our experiments. We have also analyzed the results for the default color jitter without Grayscale augmentation (CJ-). These results show that removing the Grayscale augmentation improves color sensitivity significantly. We therefore also consider this augmentation in future experiments.

## 4.3 ABLATION STUDY

Six different models were trained and evaluated with a linear classification for image classification. For resolution $32 \times 32$ the model is evaluated on CIFAR-100 and FLOWERS-102. The results in terms of accuracy are reported in Table 1 and Table 2. We identify two different trends when interpreting these results. On CIFAR-100, removing color augmentations makes the model less powerful, due to the loss of color invariance in the features extracted by the encoder. This behaviour is consistent with what was reported by Chen et al. (2020a). We see in Table 1 that if color augmentations (i.e. brightness/contrast and Random Grayscale) are removed completely (the *None* configuration), the accuracy drops by $18\%$. On FLOWERS-102 the behavior is the opposite however: removing color augmentations helps the model to better classify images, obtaining an improvement of $12.75\%$

Table 2: Accuracy results for self-supervised training on CIFAR-100 and evaluated at $32 \times 32$ on CIFAR-100 and FLOWERS-102. The reported accuracy refers to the results of the linear classifiers trained with features extracted from the different trained backbones.

| AUGMENTATION | CIFAR-100 | FLOWERS-102 |
|---|---|---|
| Default Color Jitter (CJ) | 59.93% | 30.00% |
| Default Color Jitter w/o Grayscale (CJ-) | 41.96% | 36.96% |
| Planckian Jitter (PJ) | 47.31% | 42.75% |
| LSC: [CJ, CJ-] | 62.27% | 47.65% |
| LSC: [CJ, PJ] | 63.54% | 51.66% |

of accuracy with respect to the default color jitter. This behavior confirms that color invariance negatively impacts downstream tasks where color information plays an important role.

Taking a closer look at the various augmentation on FLOWERS-102, we see that introducing more realistic color augmentations positively impacts contrastive training and produces models that achieve even better results with respect to the configuration without any kind of image color manipulation. Removing all color augmentations (None) improves results already by over 6%. Then, by simply reducing the jittering operation to influence brightness and contrast, leaving hue and saturation unchanged, yields another boost in accuracy of $5.49\%$ (to 41.96). When we start modifying chromaticity using a more realistic transformation (i.e *Planckian Jitter*), the final result is a boost of $6.28\%$ in accuracy with respect to the *None* configuration. Also, on CIFAR-100 we see an improvement of $5.38\%$ from Planckian Jitter with respect no color augmentation. Despite this improvement, in this scenario the contrastive training with the realistic augmentation does not yield better results with respect to the *Default* configuration because color only plays a minor role on this dataset.

Given the results obtained using the data augmentations reported in Table 1, and given the considerations made in Section 3.2, we evaluate the complementarity of the learned representation by combining latent spaces from different backbones. Results for two different latent space combinations are given in Table 4. On both datasets the *Latent space combination* of Default and Planckian Jitter achieves the best results. On the original CIFAR-100 task, this combination achieves a total accuracy of $63.54\%$, a $3.61\%$ improvement over *Default* and $16.23\%$ more compared to *Planckian Jitter* alone. Comparing to the LSC using the Default ColorJitter w/o Grayscale, the version with Planckian Jitter achieves a small improvement of $1.27\%$ in classification accuracy.

On the downstream FLOWERS-102 task, *Latent space combination* reaches an accuracy value of $51.66\%$: an improvement of $21.66\%$ and $8.91\%$ in accuracy respectively compared to the two original configurations. Compared to the LSC using Default ColorJitter w/o Grayscale, the combination with Planckian Jitter achieves a higher result, and a bigger gap in terms of accuracy with respect to the CIFAR-100 scenario. Here the use of Planckian Jitter brings an improvement of $4.01\%$, confirming the impact of using realistic augmentation on classification tasks for which color is important.

## 4.4 EVALUATION ON DOWNSTREAM TASKS

Given the ablation study results, we performed the analysis of the proposed configurations on other downstream tasks using the backbone trained on higher resolution images ($224 \times 224$ pixels). We report in Table 3 the results for: *Default Color Jitter*, *Planckian Jitter*, and latent space combinations.

Looking at the results, we see that the *Planckian Jitter* augmentation outperforms default color jitter on three datasets (CUB-200, T1K+, and USED). Comparing the results on FLOWERS-102 with those reported above at ($32 \times 32$) pixels, we see that default color jitter actually obtains good results. We hypothesize that for high-resolution images the shape/texture information is very discriminative, and the additional color information yields little gain (for further analysis, see also A.7). Table 3 also contains results for latent space combination, which confirm that the two learned representations are complementary. Their combination yields gains of up to 9% on T1K+. As a sanity check we also include the latent space combination of two networks separately trained with Color Jitter. This provides a small gain on some datasets, but yields significantly inferior results than LSC.

Table 3: Evaluation on downstream tasks. Self-supervised training was performed on IMAGENET at $(224 \times 224)$ and testing performed on the downstream datasets resized to $(224 \times 224)$.

| AUGMENTATION | CUB-200 | VEGFRU | T1K+ | USED | FLOWERS-102 |
|---|---|---|---|---|---|
| Default Color Jitter (CJ) | 54.52% | 67.63% | 71.44% | 59.90% | 93.16% |
| Planckian Jitter (PJ) | 56.28% | 65.84% | 77.42% | 60.03% | 90.29% |
| LSC [CJ,PJ] | **60.70%** | **74.73%** | **80.49%** | **64.07%** | **93.99%** |
| LSC [CJ,CJ] | 56.16% | 70.59% | 73.47% | 61.07% | 93.13% |
| LSC [CJ,CJ-] | 53.14% | 70.54% | 78.32% | 63.87% | 93.47% |

Table 4: Effect of Plackian Jitter on different contrastive learning models. Self-supervised training was performed on CIFAR-100 and the learned features are evaluated at $(32 \times 32)$ on CIFAR-100 and FLOWERS-102. We report the best configurations obtained on SimSiam model and retrained SimCLR and Barlow Twins with those selected configurations.

| FRAMEWORK | AUGMENTATION | CIFAR-100 | FLOWERS-102 |
|---|---|---|---|
| SimSiam | Default Color Jitter (CJ) | 59.93% | 30.00% |
| | Planckian Jitter (PJ) | 47.31% | 42.75% |
| | LSC [CJ,PJ] | 63.54% | 51.66% |
| SimCLR | Default Color Jitter (CJ) | 56.99% | 35.29% |
| | Planckian Jitter (PJ) | 47.75% | 45.00% |
| | LSC [CJ,PJ] | 61.07% | 55.78% |
| Barlow Twins | Default Color Jitter (CJ) | 56.60% | 40.78% |
| | Planckian Jitter (PJ) | 52.71% | 54.50% |
| | LSC [CJ,PJ] | 62.85% | 62.55% |
| VicReg | Default Color Jitter (CJ) | 65.23% | 49.50% |
| | Planckian Jitter (PJ) | 59.19% | 50.90% |
| | LSC [CJ,PJ] | 68.95% | 60.80% |

## 4.5 GENERALITY AND LIMITATIONS OF PLANCKIAN JITTER

To show that our approach is generally applicable to self-supervised methods exploiting color augmentations, we report in Table 4 experiments using SimCLR, Barlow Twins, and the more recent VicReg Bardes et al. (2021). Independently of the model, *Latent Space Combination* consistently achieves the best results on both datasets.

A drawback of Planckian Jitter is the quality reduction of shape and texture representations, because the extreme color transformation of the standard Color Jitter force the network to solve the contrastive learning problem mainly using shape/texture information. As we have shown, this problem can be addressed by exploiting their complimentary nature. Secondly, our current latent space combination requires the training of two separate backbones, which will also learn partially-overlapping features. A training scenario with both augmentations simultaneously in a single network while reserving part of the latent space for each augmentation could be pursued to address this limitation. Finally, object-specific augmentations that also take into account shadows, the type of reflectance, secondary light sources, inter-reflections, shadows, etc, could lead to further improvements.

## 5 CONCLUSION

Existing research on self-supervised learning mainly focuses on tasks where color is not a decisive feature, and consequently exploits data augmentation procedures that negatively affect color-sensitive tasks. We propose an alternative color data augmentation, called Planckian Jitter, that is based on the physical properties of light. Our experiments demonstrate its positive effects on a wide variety of tasks where the intrinsic color of the objects (related to their reflectance) is crucial for discrimination, while the illumination source is not. We also proposed exploiting both color and shape information by concatenating features learned with different modalities of self-supervision, leading to significant overall improvements in learned representations. Planckian Jitter can be easily incorporated into any self-supervised learning pipeline based on data augmentations, as shown by our results demonstrating improved performance for three self-supervised learning models.

## REPRODUCIBILITY STATEMENT

The code of the Planckian Jitter data augmentation procedure, written in MATLAB and PyTorch 1.7.0, is made available at `https://github.com/TheZino/PlanckianJitter`.

The training runs have been performed using Pytorch in combination with Pytorch Lightning Bolt framework, which provides an implementation of SimSiam methodology for backbone contrastive training. The model has been trained using CUDA deterministic, and random seed set to 1234.

All datasets used for the training and fine-tuning are publicly available. Only the CUB200 dataset has been pre-processed, by cropping each image using the given bounding box values, available alongside each image in the annotations files.

## ACKNOWLEDGMENTS

We acknowledge the support from the Spanish Government funding for projects PID2019-104174GB-I00, TED2021-132513B-I00, the support from the European Commission under the Horizon 2020 Programme, grant RYC2021-032765-I funded by MCIN/AEI/10.13039/501100011033 and by European Union NextGenerationEU/PRTR, and grant number 951911 (AI4Media).

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
