# OpenReview forum: "Planckian Jitter: countering the color-crippling effects of color jitter on self-supervised training"
_ICLR.cc/2023/Conference — ICLR 2023 poster_

### Official Review · Reviewer_9p9z · 2022-10-21

**Confidence:** 5
**Clarity, Quality, Novelty And Reproducibility:** The paper is lack of novelty and the …
**Correctness:** 2
**Technical Novelty And Significance:** 1
**Empirical Novelty And Significance:** 1
**Recommendation:** 3

**Strength And Weaknesses:**

* Strengths:
1. The idea is straightforward and the paper is easy to read.

* Weaknesses:
1. I am sure the suggested Planckian-based augmentation is useful for a few applications that suffer from light color changes. However, I am not sure the importance of this in general settings. The existing common color augmentation is still needed for general applications (those that are not much affected by light colors, but suffer from the limited training data). Hence, proposing only the Planckian augmentation in a whole paper is too narrow and too little in terms of the contribution.

2. A quick search on the internet shows there are a few papers proposing different color augmentation techniques for certain applications: PCA Color Augmentation, "Data-Driven Color Augmentation Techniques for Deep Skin Image Analysis", "A New Color Augmentation Method for Deep Learning Segmentation of Histological Images". However, none of these are discussed in the paper.

3. The Planckian-based augmentation might be new, yet the significance of this is minor.

4. The proposed pipeline/architecture has nothing novel.


**Summary Of The Paper:**

The paper is about color augmentation, which is sometimes used for network training. Instead of common color augmentation by changing  brightness, contrast, saturation, and hue randomly, the paper suggests Planckian-based color augmentation.

**Summary Of The Review:**

Please see my comment on novelty and quality above.

---

> ### Author Response · Authors · 2022-11-16
> **Response to Reviewer 9p9z (#1)**
>
> > **Q4.1** I am sure the suggested Planckian-based augmentation is useful for a few applications that suffer from light color changes. However, I am not sure the importance of this in general settings. The existing common color augmentation is still needed for general applications (those that are not much affected by light colors, but suffer from the limited training data). Hence, proposing only the Planckian augmentation in a whole paper is too narrow and too little in terms of the contribution.
>
> Indeed, Planckian jitter is important for applications that suffer from illuminant color changes. We think there is a wide range of potential applications that can benefit from better color features (and also suffer from illuminant changes), like for example robots navigating in indoor and outdoor environments, vision systems in fashion, food, agriculture, autonomous driving.
> Our experiments, however, showed that Planckian augmentation alone can lead to a degradation in the quality of shape representation (whereas default color jitter obtains high-quality shape features).
> For this reason, in addition to Planckian Jitter we also proposed a joint representation resulting from the two augmentations, combining the features via latent space concatenation. To this extent, our experiments demonstrate significant gains for food, birds, and a texture dataset.
> To further strengthen this point, we have expanded our experimental results with two new experiments, related respectively to a different self-supervision paradigm, and to a different dataset.
> We will add these results to any final version of this paper.
>
> 1. In addition to SimSiam, SimCLR and Barlow Twins, we now use a very recent self-supervised learning paradigm called VicReg (Bardes, A., Ponce, J., & LeCun, Y. “Vicreg: Variance-invariance-covariance regularization for self-supervised learning”  International Conference on Learning Representations, ICLR 2022), consolidating the trend of better results obtained via latent space concatenation of Planckian jitter and Default color jitter.
>
>
> |AUGMENTATION|CIFAR-100|FLOWERS-102|
> |----------------------|-------:|------:|
> | Default Color Jitter | 65.23% | 49.50% |
> | Planckian Jitter     | 59.19% | 50.90% |
> | LSC[CJ,PJ]           | 68.95% | 60.80% |
>
> 2. In addition to the existing downstream tasks, we verified the effects of our solution on scene-based recognition, where different surface colors can co-exist (USED: A Large Scale Social Event Detection Dataset, http://loki.disi.unitn.it/~used/).  The results again lead to consistent conclusions relative to the joint representation:
> | Augmentation              | "USED" dataset |
> |---------------------------|--------------------------------------|
> | Default Color Jitter (CJ) | 59.90%                               |
> | Planckian Jitter (PJ)     | 60.03%                               |
> | **LSC [CJ,PJ]**               | **64.07%**                               |
> | LSC [CJ,CJ]               | 61.07%                               |
> | LSC [CJ,CJ-]              | 63.87%                               |

---

> > ### Author Response · Authors · 2022-11-16
> > **Response to Reviewer 9p9z (#2)**
> >
> > > **Q4.2** A quick search on the internet shows there are a few papers proposing different color augmentation techniques for certain applications: PCA Color Augmentation, "Data-Driven Color Augmentation Techniques for Deep Skin Image Analysis", "A New Color Augmentation Method for Deep Learning Segmentation of Histological Images". However, none of these are discussed in the paper.
> >
> > Thank you, we will include these papers in a discussion of existing color augmentation research. We would like to also note that the mentioned papers are applied for supervised learning, in very specific application domains, whereas we propose an augmentation method for self-supervised learning and show results on several datasets representing a wide range of applications (birds, flowers, textures, food, social events).

---

> > > ### Author Response · Authors · 2022-11-16
> > > **Response to Reviewer 9p9z (#3)**
> > >
> > > > **Q4.3** The Planckian-based augmentation might be new, yet the significance of this is minor.
> > >
> > > Our paper focuses on color augmentations for the immensely popular self-supervised learning methods that have been proposed in recent years, including SimCLR (Chen
> > > et al., 2020a), MoCo (He et al., 2020), and SimSiam (Chen & He, 2021). These methods have received enormous attention from the research community (and have already been cited thousands of times). This is one of the most impactful areas of progress in artificial intelligence in recent history, as these methods allow learning representations from unlabeled data -- representations that rival those learned on supervised data. One of the crucial elements of these methods is the set of applied augmentations that are used to train them. They have in common that all are based on standard color jitter. One of the contributions of our paper is that we point out that the used color augmentation in these methods severely limits the quality of the learned color features. We therefore propose Planckian jitter as an alternative and complementary color augmentation and show in extensive experiments that indeed the combination of standard color jitter and the proposed Planckian color jitter can obtain significant performance gains in unsupervised representation learning.

---

> > > > ### Author Response · Authors · 2022-11-16
> > > > **Response to Reviewer 9p9z (#4)**
> > > >
> > > > > **Q4.4** The proposed pipeline/architecture has nothing novel.
> > > >
> > > > We would like to note that the focus of our paper is not on a new architecture. Instead, we propose a new augmentation method that can be applied to a wide range of architectures and pipelines that are used for self-supervised learning. To show this, we present results of our augmentation for several self-supervised techniques in Table 3. As seen from this table, our augmentation method can successfully be combined with SimCLR (Chen et al., 2020a), SimSiam (Chen & He, 2021), Barlow Twins (Zbontar et al. 2021), VicReg (Bardes et al. 2022).

---

### Official Review · Reviewer_kWdv · 2022-10-22

**Confidence:** 3
**Correctness:** 4
**Technical Novelty And Significance:** 3
**Empirical Novelty And Significance:** 3
**Recommendation:** 8

**Clarity, Quality, Novelty And Reproducibility:**

The paper is generally written well but there are some minor issues as listed above.

The paper presents an interesting method for color augmentation and shows its effectiveness in some scenarios.

The paper explains detailed information about the proposed method and experiments and promises to release the code.

**Strength And Weaknesses:**

Strong points.

1. A new color data augmentation method is proposed, which is demonstrated to be robust to illumination changes, useful for recognizing objects based on color information, and can be combined with other augmentation methods.

2. This work is motivated well. Figure 1 is a good visualization that illustrates the differences between the existing color jitter method and the proposed method.

3. Claims are generally supported via justifications and experiments.

Weak points.

While this paper proposes to learn color representations for tasks in which the intrinsic color of the objects is crucial, I wonder whether the proposed method can work well on 1, scene-level image-based tasks where multiple objects with different colors co-exist, and 2, Fine-grained image categorization (Sub-category recognition).

The writing could be improved for better readability. Some issues are listed below.

1. The paper does not use hyperlinks for figures/tables/references discussed in the main text, making it difficult to locate them or refer to the references for more information.
2. Figure 2 does not have "p_2" on it.
3. It is quite strange to suddenly describe the proposed network architecture in the related work section.
4. The "Data augmentation" paragraph of Section 2 says "x_1, x_2 used in the ... in Eq. 2" but Eq. 2 does not contain "x_1" and "x_2".
5. Figure 4 is mentioned before Figure 3 in the main text.

**Summary Of The Paper:**

This paper proposes a physics-based color data augmentation method (called Planckian Jitter), to train deep models to be robust to illumination changes. Experiments show that this method can produce a representation that is complementary to the representations learned by current color jitter augmentation methods, which further boosts performances on a variety of downstream tasks. The paper also presents an analysis of the color sensitivity of deep models to training strategies. Experiments show that the proposed method is robust to illuminant changes and can be combined with existing self-supervised learning methods to further boost their performance.

**Summary Of The Review:**

The paper presents an interesting method for color augmentation and shows its effectiveness in some scenarios. I do not find major issues but am concerned about its further applications. I hope the authors could discuss them, which should be inspiring for future research.

---

> ### Author Response · Authors · 2022-11-16
> **Response to Reviewer kWdv (#1)**
>
> > **Q3.0** While this paper proposes to learn color representations for tasks in which the intrinsic color of the objects is crucial, I wonder whether the proposed method can work well on 1, scene-level image-based tasks where  with different colors co-exist, and 2, Fine-grained image categorization (Sub-category recognition).
>
> Thank you for these suggestions.
>
> Following the suggestion to check the results for scene-based recognition where different colors can co-exist, we performed an additional experiment with USED: A Large Scale Social Event Detection Dataset (http://loki.disi.unitn.it/~used/). The dataset consists of fourteen different categories of social events from around the world. In this dataset images do not contain a single object that dominates the classification, but rather the interaction between multiple objects and the background scene are important to make correct predictions. We took 1K images/class for training, and 500 images/class for testing. The results for linear classification are:
> | Augmentation              | "USED" dataset |
> |---------------------------|--------------------------------------|
> | Default Color Jitter (CJ) | 59.90%                               |
> | Planckian Jitter (PJ)     | 60.03%                               |
> | **LSC [CJ,PJ]**               | **64.07%**                               |
> | LSC [CJ,CJ]               | 61.07%                               |
> | LSC [CJ,CJ-]              | 63.87%                               |
>
> These are consistent with the preexisting experiments.
>
> With respect to fine-grained image categorization and sub-category recognition, we realize that we were not sufficiently clear in describing our choice of datasets: the VegFru dataset is indeed hierarchically-structured, with 200 sub-categories belonging to 15 super categories. This characteristic will be made explicitly clear in any final version of the paper. Results related to VegFru are presented in Table 3 and show that the joint application of Planckian and default Color Jitter via Latent Space Combination produces the overall best results.

---

> > ### Author Response · Authors · 2022-11-16
> > **Response to Reviewer kWdv (#2)**
> >
> > > **Q3.1** The paper does not use hyperlinks for figures/tables/references discussed in the main text, making it difficult to locate them or refer to the references for more information.
> >
> > We apologize for the inconvenience. We understand that this is a known issue with the internal PDF conversion system of OpenReview, which in some cases strips the document of all hyperlinks. This issue will be solved in any final submission.
> >
> > > **Q3.2** Figure 2 does not have "p_2" on it.
> >
> > The process described in Section 2 is alternated between the two image versions, however only one application is shown in Figure 2 for brevity (with MLP applied to $z_1$, therefore missing $p_1$).
> > This will be made explicit in the text as:
> >
> > “The MLP is applied in alternation to either $z_1$ or $z_2$, producing respectively $p_1$ or $p_2$. To this extent, Figure 2 only shows an instance for $x_1$, therefore missing $p_2$.”
> >
> > > **Q3.3** It is quite strange to suddenly describe the proposed network architecture in the related work section.
> >
> > Due to space limitations, we opted to integrate in Section 2 the description of existing solutions for self supervised learning, such as SimSiam, with nomenclature specific from our proposal, namely Planckian jitter. The core of our physics-inspired data augmentation is instead introduced later on in a dedicated Section 3.
> >
> > > **Q3.4** The "Data augmentation" paragraph of Section 2 says "x_1, x_2 used in the ... in Eq. 2" but Eq. 2 does not contain "x_1" and "x_2".
> >
> > We thank the reviewer for finding out this typo.
> > We will fix the sentence as reported:
> >
> > “These operations are randomly applied to an image to generate the different views x_1, x_2
> > **from which are extracted the features $z_1$ and $z_2$**
> > used in the self-supervision loss in Eq.2.”
> >
> > > **Q3.5** Figure 4 is mentioned before Figure 3 in the main text.
> >
> > We note that Figure 4 belongs to the appendix, whereas Figure 3 belongs to the main text. We included a direct reference to a figure in the appendix to facilitate reading, but we will correct it if this violates the conference format.

---

### Official Review · Reviewer_w1qF · 2022-10-24

**Confidence:** 4
**Correctness:** 3
**Technical Novelty And Significance:** 3
**Empirical Novelty And Significance:** 3
**Recommendation:** 6

**Clarity, Quality, Novelty And Reproducibility:**

The representation is clear, the quality can be improved, the novelty is good, and the reproducibility of this work seems to be good.

**Strength And Weaknesses:**

Strength
+ Color is important in image classification, especially for some fine-grained image classification, such as flower classification.
+ Considering the physical and realistic procedure for color data augmentation is novel.

Weakness
- The proposed method assumes that the images are captured under the daylight D65. However, in the daily life, we capture image under various lights, such as LED light and other daylight D55 and so on.
- The improvement of proposed method is not significant in some cases. According to Table 1, the proposed Planckian Jitter performs significantly worse than default color jitter on CIFAR-100 dataset. Besides, Planckian Jitter without brightness and contrast (B&C) or Planckian-based chromaticity performs even worse than without data augmentation.
- The compared methods are not comprehensive. The proposed Planckian Jitter mainly performs on SimSiam (2021), and generally performs on SimCLR (2020) and Barlow Twins (2021). It is better to perform on more recently proposed contrastive learning methods. Besides, the goal of color data augmentation is to improve the performance of image classification network. Thus, why not directly perform the proposed method on advanced image classification network, such as ResNet-101, Swin-Transformer and so on.
- The selection of some parameters is difficult to understand, such as C_B, C_C in Equation 5. The effectiveness of these parameters is unknown.
- There are some typos. For examples, ‘We’ of term 4 in page 4.


**Summary Of The Paper:**

This paper proposes a color data augmentation method for self-supervised learning or contrastive learning, which applies physically illuminant variation to images and the illuminants described by Planck’s Law for black-body radiation. The experimental results show that the proposed color augmentation method outperforms existing methods in some cases.

**Summary Of The Review:**

Applying physical and realistic color data augmentation to network training is valuable, but the signification, comparison and some other aspects of this work can be improved.

---

> ### Author Response · Authors · 2022-11-16
> **Response to Reviewer w1qF (#1)**
>
> > **Q2.1** The proposed method assumes that the images are captured under the daylight D65. However, in the daily life, we capture image under various lights, such as LED light and other daylight D55 and so on.
>
> The D65 reference white, used for our color space conversions, means that a neutral surface illuminated by average daylight conditions would appear achromatic: this assumption allows us to interpret our augmentation as introducing a controlled temperature variation into a neutrally-lit scene. In general, the images from computer vision datasets are assumed to be acquired with some form of camera automatic white balance, therefore supporting this interpretation.
>
> In a case where the initial image is not white balanced, the application of planckian jitter will still work as introducing a natural color cast, which is roughly laying on the Planckian locus if the initial illuminant is extracted from that same distribution.
> L.E.D. sources are an additional possibility for further exploration. At the moment, however, we take note that highly saturated sources are relatively less common, as shown by Buzzelli et al. 2022 in reference to green/magenta lights (Analysis of biases in automatic white balance datasets and methods. Color Research & Application.), and introducing them with a proportional sampling probability would result in extremely sparse occurrences.
>
> A possible further step to enforce a tighter control on the applied augmentation would be to first preprocess the image with automatic white balance algorithms.

---

> > ### Author Response · Authors · 2022-11-16
> > **Response to Reviewer w1qF (#2)**
> >
> > > **Q2.2** The improvement of proposed method is not significant in some cases. According to Table 1, the proposed Planckian Jitter performs significantly worse than default color jitter on CIFAR-100 dataset.
> > Besides, Planckian Jitter without brightness and contrast (B&C) or Planckian-based chromaticity performs even worse than without data augmentation.
> >
> > We agree with the reviewer about the fact that the proposed augmentation does not bring improvement in all the cases. We aim for a representation that is complementary to the representation learned with standard color jitter, so that, when combined, they obtain superior performance.
> >
> > As we reported in section 4.3, our thought on these results is that the effectiveness of the proposed augmentation is related to the importance of the color information for the discrimination process during classification. In fact, as shown in Table 1 and Table 3, when moving to other kind of datasets where color information covers a more important role for the class discrimination (such as the Flowers-102 or the T1K+), the Planckian Jitter brings an improvement in the final classification accuracy, with respect to the Default Color Jitter augmentation.
> >
> > In the case of datasets such as the CIFAR-100, structure plays a bigger role in the classification task. As we pointed out in Section 4.5, a drawback of the Planckian Jitter is the reduction in the quality of the shape and texture representations: “The extreme color transformation of the standard Color Jitter forces the network to solve the contrastive learning problem mainly using shape/texture information”. This behavior has also been analyzed in Section 4.2, where we showed how the model trained using the Planckian Jitter is much more color sensitive than its counterpart trained with the default Color Jitter augmentation. The removal of the brightness and contrast terms from the Planckian Jitter augmentation reduces even more the quality of the shape/texture representation from the final model while improving the color sensitivity of the model, explaining why the results on CIFAR-100 became even worse with respect to the full version of the Planckian Jitter.
> > The proposed Latent Space Combination aims to combine the benefits from the two different representations, showing in the comparisons in Table 2 that the final accuracy actually benefits from this type of combination. This is further confirmed by the results in Table 3 where the combination always obtains the best results.

---

> > > ### Author Response · Authors · 2022-11-16
> > > **Response to Reviewer w1qF (#3)**
> > >
> > > > **Q2.3 a)** The compared methods are not comprehensive. The proposed Planckian Jitter mainly performs on SimSiam (2021), and generally performs on SimCLR (2020) and Barlow Twins (2021). It is better to perform on more recently proposed contrastive learning methods.
> > >
> > > We now also run the Planckian Jitter with the more modern VicReg (Bardes, A., Ponce, J., & LeCun, Y. “Vicreg: Variance-invariance-covariance regularization for self-supervised learning”  International Conference on Learning Representations, ICLR 2022). The results are provided in the following Table. As can be seen, the same trends as for the other self-supervised methods can be observed: again a significant performance gain is observed when combining both standard color jitter with Planckian color jitter. We can also observe that this newer method obtains overall better results for all evaluated representations.
> > >
> > > |AUGMENTATION|CIFAR-100|FLOWERS-102|
> > > |----------------------|-------:|------:|
> > > | Default Color Jitter | 65.23% | 49.5% |
> > > | Planckian Jitter     | 59.19% | 50.9% |
> > > | LSC[CJ,PJ]           | 68.95% | 60.8% |
> > >
> > > > **Q2.3 b)** Besides, the goal of color data augmentation is to improve the performance of image classification network. Thus, why not directly perform the proposed method on advanced image classification network, such as ResNet-101, Swin-Transformer and so on.
> > >
> > > In our work, we specifically target data augmentation for self-supervised learning.
> > > Self-supervised learning aims to map two different views (computed by taking two different data augmentations of the same image) to the same point in latent space.
> > > It was found that color jitter is a very important augmentation to obtain good results: in “Simple Framework for Contrastive Learning of Visual Representations” (Chen et al. 2020) the authors discuss Fig. 5 of their paper, reasoning that “it is critical to compose cropping with color distortion in order to learn generalizable features.” However, what was not well understood is that this led to a significant degradation of the quality of the learned color representations (the topic of our paper). In the next section (Section 3.2) the same authors write “When training supervised models with the same set of augmentations, we observe that stronger color augmentation does not improve or even hurts their performance”.
> > > We agree that it would be interesting to see if Planckian Jitter also has this detrimental effect in the supervised setting, but because we focus on self-supervised learning in this paper, we have not included those results and will consider them as future work.
> > >
> > > Even though color augmentations are sometimes used in supervised image classification network learning, they play a much less important role than for self-supervised learning (Shorten  & Khoshgoftaar (2019). A survey on image data augmentation for deep learning. Journal of big data). Nevertheless, we agree that it would be interesting to evaluate Planckian jitter for supervised learning, and we plan to do this for future work.

---

> > > > ### Author Response · Authors · 2022-11-16
> > > > **Response to Reviewer w1qF (#4)**
> > > >
> > > > > **Q2.4** The selection of some parameters is difficult to understand, such as C_B, C_C in Equation 5. The effectiveness of these parameters is unknown.
> > > >
> > > > We referred to a standard implementation of SimSiam as default parametrization for Contrast and Brightness augmentations:
> > > > * https://github.com/Lightning-AI/lightning-bolts/blob/c26c8d8f407de386038d5fb13297233a8aa052e7/pl_bolts/models/self_supervised/simclr/transforms.py#L43
> > > > * https://pytorch.org/vision/0.14/generated/torchvision.transforms.ColorJitter.html
> > > >
> > > > Given our main focus on redesigning the chromaticity component of data augmentation, we instead leave the hyperparameter optimization of other components as a potential direction for future works.
> > > >
> > > >
> > > > Finally, we also further revised the whole paper for typos.

---

### Official Review · Reviewer_HQBn · 2022-10-25

**Confidence:** 4
**Correctness:** 4
**Technical Novelty And Significance:** 3
**Empirical Novelty And Significance:** 3
**Recommendation:** 8

**Clarity, Quality, Novelty And Reproducibility:**

The paper is clearly written and should have enough detail to reproduce the work. The analysis is original and asks an important question about something that is usually overlooked in network training.

**Strength And Weaknesses:**

The strength of the paper is the intentional analysis of the color augmentation process used in deep network training. In particular, the author's hypothesize that the standard color augmentation process strongly reduces color as a useful feature for classification. While for some objects, color is not strongly indicative (though certain distributions may be), for other tasks, such as natural objects like flowers and birds, color may be a critical feature. The improved performance on the flowers database appears to support this hypothesis. Likewise, the evaluation on two other data sets shows that the embedding trained with the Planckian illuminants is robust across simulated illumination changes, while the embedding trained with traditional color augmentation is not.

There are other potential illuminants besides Planckian, but for the purpose of analyzing this question, it is a reasonable model to choose.

This is only a first step towards appropriate training using color augmentation. The whole image augmentation model means that there is no selectivity by surface or current illumination conditions. An object in shadow, for example, is in a different lighting condition than the same object that is fully lit, and simulating the spectral rotation of a new direct illuminant shouldn't actually modify the shadowed areas as  much or at all.  Likewise, there is still a need for color augmentation of objects for which color is not discriminative.  However, the use of the combined embedding acknowledges that. It could be more explicit in the paper that there are several valid types of color augmentation, and that whole image color augmentation is a pretty heavy hammer compared to an ideal selective augmentation process.

Given the need to train two separate embeddings, one for each type of jitter, how does this get implemented in standard network architectures?  Does it have utility beyond this particular network structure?

**Summary Of The Paper:**

The paper examines the use of color in data augmentation for training deep networks and how it impacts the embedding representation of the input. In particular, they compare the traditional random color augmentation strategy with a color augmentation based on simulating changing illuminants. The illuminants used are drawn from the distribution of Planckian illuminants. The authors train the network in an unsupervised manner to learn an embedding space and then build a linear classifier using the embedding to test the effectiveness of the embedding on two standard data sets. Using a combined embedding that makes use of a network trained using the traditional color augmentation and a network trained using the Planckian color augmentation provides the best performance.

**Summary Of The Review:**

This is an important issue in network training. However, it might be challenging to integrate it into a standard architecture/classification problem given the utility of concatenating separate embeddings based trained on different types of color jitter.

---

> ### Author Response · Authors · 2022-11-16
> **Response to Reviewer HQBn (#1)**
>
> > **Q1.1** This is only a first step towards appropriate training using color augmentation. The whole image augmentation model means that there is no selectivity by surface or current illumination conditions. An object in shadow, for example, is in a different lighting condition than the same object that is fully lit, and simulating the spectral rotation of a new direct illuminant shouldn't actually modify the shadowed areas as much or at all.
> Likewise, there is still a need for color augmentation of objects for which color is not discriminative. However, using the combined embedding acknowledges that. It could be more explicit in the paper that there are several valid types of color augmentation, and that whole image color augmentation is a pretty heavy hammer compared to an ideal selective augmentation process.
>
> We agree with the reviewer that our proposal can be seen as a first step towards more realistic color augmentations, and that there is still a long way to go to achieve truly realistic color augmentations. As mentioned by the reviewer, object-specific augmentations that also take into account shadows, the type of reflectance (e.g. Lambertian or specular), secondary light sources, inter-reflections, shadows etc, could lead to further improvements.
>
> The proposed Planckian augmentation allows the network to learn higher quality color features, however, since this simplifies the task for the contrastive learner, the learned representation more often relies on color information and thus learns lower quality shape/texture features. We therefore combine both to obtain a representation that has both high-quality color and high-quality shape features.
>
> We will incorporate a discussion of this in any final version of the paper.

---

> > ### Author Response · Authors · 2022-11-16
> > **Response to Reviewer HQBn (#2)**
> >
> > > **Q1.2** Given the need to train two separate embeddings, one for each type of jitter, how does this get implemented in standard network architectures? Does it have utility beyond this particular network structure?
> >
> > In practice, one would just apply the two trained backbones to images and concatenate the representations. The proposed method does not impose restrictions on the backbone architecture. In the literature, these self-supervised learning strategies have been applied to a variety of architectures, including for example Vgg, ResNet, WideResNet, MobileNet, and ShuffleNet (F. Ding, Multi-level Knowledge Distillation via Knowledge Alignment and Correlation, 2022, B.Yildiz, AmsterTime: A Visual Place Recognition Benchmark Dataset for Severe Domain Shift, 2022).
> >
> > It is expected that these representations actually share part of the features. A network which would be able to generate both representations (based on default jitter and on Planckian jitter) would be more efficient (requiring only a single network and yielding a more compact image representation than the one obtained by concatenation). We are considering learning a single network trained jointly with both Planckian and standard color augmentations (restricting the losses to only operate on part of the latent space) as a promising future research direction.

---

### Author Response · Authors · 2022-11-18
**Rebuttal revision uploaded**

We would like to thank all reviewers and the chairs for their work on our manuscript.

We have addressed all raised issues, and we believe that this significantly helped in improving the quality of our submission.

The revised version is available for download.


Thank you,

the authors.

---

### Decision · Program_Chairs · 2023-01-20

**Decision:**

Accept: poster

**Justification For Why Not Higher Score:**

While useful, the paper's contribution is somewhat slight - a poster will be sufficient to expose the idea to other practitioners.

**Justification For Why Not Lower Score:**

Reviewers are in agreement that it is timely to publish this work, and that ICLR is an appropriate venue.

**Metareview: Summary, Strengths And Weaknesses:**

The paper re-examines color in the context of data augmentation.  The new augmentation is shown to have beneficial effects in some applications, and to additively combine with existing augmentations in others.

Reviewers concur: the paper's proposed augmentation is novel, and will be useful in some scenarios e.g. fine grained recognition.  The idea is simple, but reviewers consider it worthy of exposure to the community, so lean towards acceptance.

R1 expresses concern that table 1 shows the augmentation is not useful in general, but other reviewers, and the AC are convinced by table 3, showing that it is generally useful when combined with other augmentations.

Reviewers were concerned that the experiments were shown only for simple models, and that the augmentation might not help with more complex models, but the rebuttal's inclusion of the VicReg experiment mitigated this concern.


**Note From Pc:**

if the above contains the word "oral" or "spotlight" please see: "oral" presentation means -> notable-top-5% and "spotlight" means -> notable-top-25%. As stated in our emails, we are disassociating presentation type from AC recommendations

**Summary Of Ac-Reviewer Meeting:**

Reviewers and AC met to discuss the paper.  Topics of discussion were

- R1's concern about the general applicability of the technique.  Other reviewers agreed that (a) it is shown to be useful in conjunction with other techniques, and (b) the value of re-examining color augmentation was a contribution in itself.

- Application under other lighting conditions.  It was agreed that the rebuttal made a good case for the general utility of the daylight-only analysis, and that it might be future work to consider other illuminations for special-case applications.

- The idea is of course simple, but that in itself may be seen as much as a strength than a weakness